# A new shortcut for competitive sports development? The purpose of and strategy for developing and introducing sepaktakraw in Taiwan

**Ding-Yi Wu**[1], **Tien-Chin Tan**[2]*, **Yan-Ting Wang**[1]*

**1** The School of Physical Education, Quanzhou Normal University, Quanzhou, China, **2** Graduate Institute of Sports and Leisure Management, National Taiwan Normal University, Taipei City, Taiwan

* tantony60@gmail.com (TCT); 15959962287@163.com (YTW)

**Data Availability Statement:** All relevant data are within the paper.

**Funding:** The author(s) received no specific funding for this work.

## Abstract

A growing number of countries and regions have introduced emerging sports in recent years; however, few studies have been conducted to determine how and why they do so. This article focuses on why a new sport, sepaktakraw, was introduced in Taiwan and how strategies for achieving international sporting success were developed in a short period. The sports policy factors leading to the international sporting success model proposed by De Bosscher et al. were adopted as this study's analytical framework. Data were obtained from official government and sport federation documents, media reports, and semistructured interviews with 18 key stakeholders. This study revealed three main reasons for the introduction of sepaktakraw in Taiwan: (1) low introduction costs, (2) breakthroughs to win medals at the Asian Games, and (3) advantageous entry into international sports organization leadership. Seven aspects are prioritized in Taiwan's sepaktakraw development strategy: (1) obtaining government financial support; (2) institutionalizing organisational decision-making; (3) taking universities as a breakthrough point and then gradually shifting to promote popularization at the grassroots level; (4) selecting players on the basis of competition results; (5) using school badminton courts; (6) strengthening coach training; and (7) actively participating in and striving to host international tournaments. The findings could provide guidelines and examples for other countries or regions to follow when introducing new sports and moving towards the successful development of competitive sports.

## Introduction

The sport of sepaktakraw is prevalent and is often played recreationally or competitively on the streets in Southeast Asia. During the game, the feet, head, knees, and shoulders are used instead of the hands. Once the ball is served over the net, players use their feet to attack more efficiently and, therefore, often appear to perform 360-degree spins to hit the ball. Moreover, because the rules are similar to those of volleyball, countries outside Southeast Asia often call sepaktakraw "foot volleyball" [1].

**Competing interests:** The authors have declared that no competing interests exist.

Today, sepaktakraw is an internationally competitive sport. It made its first international appearance in 1965 at the Kuala Lumpur Southeast Asian Games, officially entered the Asian Games in 1990, and was included as one of the official disciplines in the 2008 Asian Beach Games [2]. Although sepaktakraw has not yet been included in the Olympic Games, its inclusion as an official sport in the 1990 Asian Games led many Asian countries, including many small countries or regions such as Taiwan, to introduce the sport. Why did these countries decide to introduce a new competitive sport, and how can they develop new sports?

Despite the proliferation of research on elite sports, there is little research in the English literature about new elite sport pathways in small countries or regions. The potential for small countries or regions to win more medals, which is underexplored, is not commensurate with the increasing significance of new elite sports. Several studies have examined public policy solutions to various social problems, which is an institutionalized approach for solving relevant and real-world problems [3–11]. For example, Sun et al. analysed the complex relationship between the digital economy and carbon emissions from a public policy perspective and provided actionable policy recommendations [6]. These policy-related studies have been impactful. However, applying public policy perspectives to elite sport development in small countries or regions is rare. These two gaps are addressed in the present study.

Accordingly, this study explored why Taiwan introduced a new sport, sepaktakraw, and the strategies adopted to rapidly achieve international sporting success. The sports policy factors leading to international sporting success (SPLISS) model proposed by De Bosscher in 2006 was the main theoretical framework of this study [12]. This study investigated the reasons for the introduction of sepaktakraw to Taiwan and its developmental strategies, which can provide a reference for the subsequent successful development of this sport in Taiwan. In addition, the results of this study provide other countries and regions with a reference for introducing new sports in the future and demonstrate how to successfully develop competitive sports. Fig 1 presents the framework of the research content of this paper.

The remainder of the paper comprises five sections: literature review, methods, findings, discussion and conclusion.

## Literature review

Four theoretical studies of competitive sport policies that contribute to the success of international sports are systematic and frequently cited and discussed: Oakly and Green [13], Digel [14], Green and Houlihan [15], and De Bosscher et al. [12]. Green and Oakly systematically studied competitive sports performance in the United States, Spain, France, and Australia. The authors summarized ten critical factors for the successful development of competitive sports. Among these studies, that of De Bosscher et al. is the most representative [12]. The team analysed the results of other research teams and extensively reviewed the literature on how meso-level policies affect international sports performance. The team then summarized the nine policy pillars of the SPLISS model. Recently, international scholars have widely used the SPLISS model to examine the development of sports in a specific country or the link between national sports policies and international sports performance [16–20]. This model has become one of the leading analytical models in competitive sports research from a public policy perspective [21].

The SPLISS model is composed of nine pillars (see Table 1). Pillar 1 involves financial support, in addition to government funding, as well as diversified funding channels such as donations from nongovernmental organizations. Pillar 2 is an integrated approach to policy development that involves smooth communication between organizations, simplifying the overall organisational structure, and a clear division of labour. Pillar 3 involves the foundation

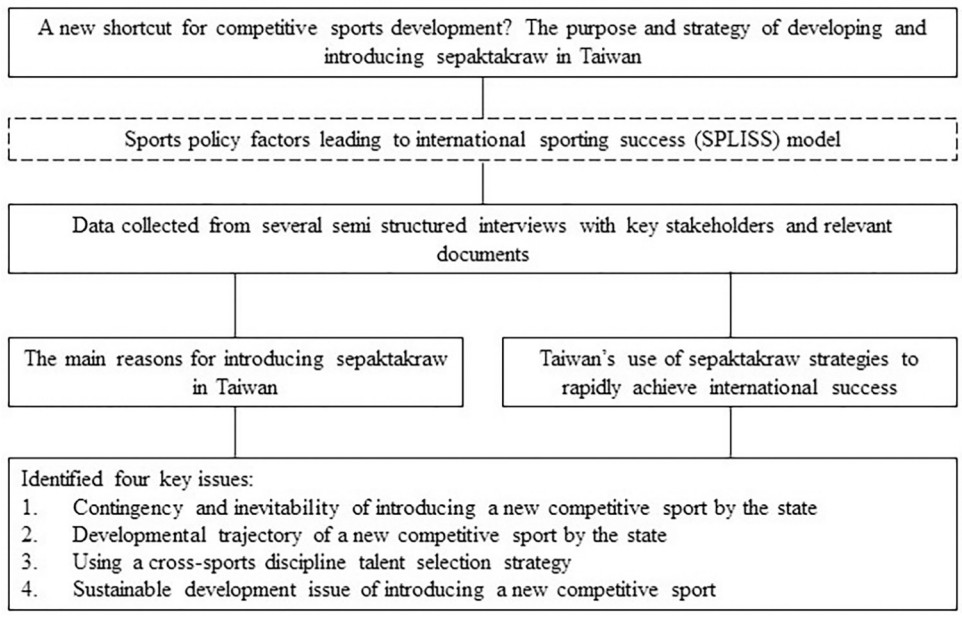

**Fig 1. Framework diagram of the paper.**

and participation, for example, in the education system, which requires an adequate athletic population at all levels. Pillar 4 is an identification and development system necessary for selecting and training talented individuals after selection. Pillar 5 is athletic and postcareer support needed for funding athletes' livelihoods, training opportunities, and career development. Pillar 6 includes training facilities, including accessible, high-quality facilities for competitive and noncompetitive sports. Pillar 7 entails coaching provision and coach development, which requires sufficient and high-quality coaches and systems to support coach development. Pillar 8 includes the opportunities to participate in international competitions, organize international competitions, and participate in national competitions. Finally, Pillar 9 describes the scientific research needed for the full development of coaching, sports science, and sports medicine.

**Table 1. The nine pillars of the SPLISS model.**

| Order | Pillars | Key content |
|---|---|---|
| 1 | Financial support | • Diversified funding channels |
| 2 | Integrated approach to policy development | • An efficient communication process between organizations<br>• Simplifying the overall structure of the organization<br>• A clearly defined division of labour◎ |
| 3 | Foundation and participation | • Sufficient athletes at all levels |
| 4 | Identification and development talents system | • Identifying and training talented individuals once they have been selected |
| 5 | Athletic and postcareer support | • Providing funding for athletes' livelihoods<br>• Providing career development assistance |
| 6 | Training facilities | • High-quality, accessible facilities for competitive purposes |
| 7 | Coaching provision and coach development | • Sufficient numbers of and high-quality coaches<br>• A system for supporting the development of coaches |
| 8 | (Inter)national competition | • Participation in international or national competitions<br>• Organization of international competitions |
| 9 | Scientific research | • Application of scientific research in a comprehensive manner |

Compared with other analytical frameworks for exploring policies in competitive sports, such as Oakly and Green [13], Digel [14], and Green and Houlihan [15], the SPLISS model De Bosscher et al. has several advantages. Lin and Tan [22] noted that the factors in the SPLISS model are more specific and explicit than those proposed by Oakley and Green [13]. Factors such as scientific research and coach training and development, for example, are explicit about the details of success in competitive sports. Lee et al. [23] noted the many advantages of the SPLISS model, including a hierarchical system of summarizing the literature into constructs and subtopics. In addition, the authors conducted a policy process analysis of policy inputs to medal outputs. Despite its many advantages, the SPLISS model has several limitations. For example, it is becoming increasingly important for competitive sports to consider cultural factors, as there are always cultural differences across nations. Therefore, the SPLISS model focuses more on country-by-country comparative analyses of policy factors [24]. According to De Bosscher, the author of the SPLISS model, using quantitative indicators that look only at the pillar's total score without examining its construction, could send misleading policy messages or lead to misinterpretation. The interaction between elite policy and social, economic, and cultural contexts may also be overlooked. Consequently, qualitative information is needed to establish a more general conclusion [24]. Furthermore, Henry et al. noted six problems associated with SPLISS and variable-oriented approaches to comparing sporting nations, including philosophical assumptions and causal variables, the black box problem, issues of internal validity, nonequivalence and reliability, neglect of agency, and misconceptions about mixed methods [25]. Although the model is not without its flaws, it still has many advantages, such as a hierarchical index structure. Moreover, scholars have commonly used it to analyse competitive sport development [17–20, 26–28]. Thus, compared with other frameworks for analysing competitive sports development, it is appropriate.

Various researchers have adapted and used the SPLISS framework to study elite sports development in general and in specific contexts to gain empirical insights into each pillar. Chen et al. [26] and Tan and Mick [27] noted that the successful development of competitive sports in both Korea and China has been attributed to diversified funding channels, simplified administrative organization, clear departmental responsibilities, increased participation in competitive sports through the education system, systematic selection methods and the monitoring of competitive athletes. Truyens [28] reported that in most countries/regions with competitive athletics, financial and programmatic support is tailored to the specific performance of competitive athletes. Some competitive track and field athletes receive additional governmental or commercial support. Moreover, Green and Houlihan [15] argued that competitive training and competition facilities, as well as sports science institutes or universities, can be shared with other sports and have been recognized as significantly contributing to the successful development of competitive sports. With respect to the development of competitive cycling in China, the need for a greater quantity and quality of coaches in the early years was one of the problems that hindered the sport's successful development [29]. Finally, Sotiriadou et al. [19] suggested that one of the critical strategies for successfully developing competitive canoeing in Australia was the availability of well-developed sports science research, such as biomechanics and psychology, nutrition, strength and conditioning, athlete training load tracking, massage and physiotherapy treatments, and effective management and monitoring.

The application of the SPLISS model by scholars in the past has focused on comparing competitive sports policy development in various countries via the SPLISS model; however, the following scholars have slowly transitioned to a more diversified application. Reis et al. [30] used the SPLSS model to organize and analyse judo science articles catalogued in the Web of Science. Venegas-Yazigi et al. [31] defined athletic funding via the SPLISS model's funding support pillar. Furthermore, they analysed the relationship between funding and international

sport performance among Chilean athletes. Sport-specific policies, including archery in Korea [18], canoeing in Australia [19], and judo in Japan [32], are explored by other groups via the SPLISS model. Most of these studies, however, focus on official Olympic sports and examine the distance between flourishing and the success of competitive sports in various countries. This study examines the development of an emerging competitive sport, sepaktakraw, from the introduction of the sport to its gradual establishment in a new country or region. This study also examines the specific actions that should be taken for an emerging sport to be developed as a sustainable or successful competitive sport in a new country or region. Therefore, exploring the development of sepaktakraw in Taiwan via the SPLISS model to understand the reasons for the introduction of an emerging competitive sport to a small country or region and how better international sports performance can be achieved in a short period is important and should provide value to relevant academic and applied groups.

The SPLISS model has been widely used to explore the development of competitive sports and sport-specific policies in different countries, identifying the factors that may make it easier for a country to succeed in developing competitive sports. Therefore, the nine pillars of the SPLISS model were used as the basis of this case study to examine the developmental process of sepaktakraw in Taiwan, from the sport's introduction to its gradual establishment. The aspects of the SPLISS model that have been emphasized in the development strategy and which aspects have not yet been achieved are analysed. This study's methodology is described in the next section.

## Methodology

This study examined how a sport (sepaktakraw) was developed from scratch and introduced in a new country/region (Taiwan) through a single case study approach [33], which was chosen for three reasons. First, sepaktakraw is an exotic and non-Taiwanese local sport discipline [1, 2]. Second, sepaktakraw was recently introduced to Taiwan, and good results have been achieved in international competitions [34]. Third, the relevant data can be obtained from archives, websites, and interviews. The above reasons align with Denscombe's principles of case selection considerations, including specific significant attributes, convenience, and feasibility [35].

### Data collection and analysis

To understand the development of competitive sepaktakraw in Taiwan, this study collected data from domestic and foreign journal articles, media reports, official documents, and meeting records from the Taiwan government's sports department and the Chinese Taipei Sepak-Takraw Federation (CTSTF).

The relevant data were also collected through semistructured interviews conducted during two periods: from March to April 2016 and from December 2022 to March 2023. The first interview period was selected because there had been initial developments by this time since Taiwan introduced the sport in 2008, such as the establishment of representative teams and participation in international tournaments, as well as initial results in international tournaments. Moreover, the same group of people were still involved in developing sports in Taiwan. The consistency of these participants can help us better grasp the development of sepaktakraw in Taiwan from the introduction of the sport to its gradual establishment in 2008–2016. The second reason for choosing the interview period is that sepaktakraw in Taiwan has developed quite significantly in a fairly short period of 7 years. Taiwan has achieved good success in international sepaktakraw tournaments. Moreover, the development of sepaktakraw in Taiwan was gradually managed by different people [34]. By interviewing the stakeholders involved in

sepaktakraw in Taiwan, further insights into the path from its gradual establishment to its flourishing can be gained. Therefore, interviews were conducted at two different time points to explore how Taiwan's sepaktakraw developed out of nothing. This approach also explores the recent changes in its rapid development. Researchers and practitioners may find these findings useful and worthy of attention. In this study, an interview outline was developed according to the theoretical framework of the SPLISS model. Key stakeholders involved in the development of sepaktakraw in Taiwan were recruited through purposive and snowball sampling to understand the sport's introduction and subsequent development. The stakeholders' selection criteria were that they were first involved in sepaktakraw development in Taiwan or that they had an important position in its development. The eighteen interviewees included one official from the Sports Administration, one official from the Ministry of Education (SAME), two senior administrators, two previous and current CTSTF directors and presidents, six previous and current national team coaches and players, and two separate novel promoters. Notably, the 2016 respondents, apart from an official from SAME, were among the first to be involved in sepaktakraw development in Taiwan. Furthermore, three interviewees in 2022 and 2023 had experience coaching and playing representative teams. The interviewees of the CTSTF include those involved in introducing sepaktakraw to Taiwan (the former CTSTF chairperson), former team coaches who continue to lead teams in middle schools to participate in national competitions, and those who continue to have business dealings with the current CTSTF (the former CTSTF director). The CTSTF interviewees included the person most responsible for introducing sepaktakraw to Taiwan (former CTSTF President), former delegation team coaches who continue to lead high school teams in attending national sepaktakraw tournaments, and former CTSTF executives who continue to have business relationships with the CTSTF. In short, the vast majority of the respondents in this study witnessed the development of sepaktakraw in Taiwan. The list of interviewees included in this study is shown in Table 2.

**Table 2. The list of interviewees.**

| Interviewee code | Service unit and title | Date of interview |
|---|---|---|
| A | Official from SAME | April 28, 2016 |
| B | Administrator of the CTSTF | March 2, 2016 |
| C | Administrator of the CTSTF | March 17, 2016 |
| D | Director of the CTSTF | March 17, 2016 |
| E | National team coach | March 18, 2016 |
| F | National team coach | April 23, 2016 |
| G | National team player | March 4, 2016 |
| H | National team player | March 2, 2016 |
| J | Promoter of sepaktakraw | March 8, 2016 |
| K | Promoter of sepaktakraw | March 8, 2016 |
| L | National team coach | December 31, 2022 |
| M | National team coach | January 16, 2023 |
| N | National team player | January 14, 2023 |
| O | National team coach | February 25, 2023 |
| P | Former director of CTSTF | February 28, 2023 |
| Q | Former national team coach | March 1, 2023 |
| R | Director of the CTSTF | March 1, 2023 |
| S | Former president of CTSTF | March 31, 2023 |

We interviewed these individuals in training halls, federation offices, and competition venues. Before each interview, we sent the interviewee a consent form to seek permission to conduct the interview and obtain consent to record the process. After the verbatim transcripts of the interviews were compiled, we also sent them back to the interviewees to assist in confirming and filling in missing content.

This study used a qualitative content analysis method, where all the data, including those from the documents and interviews, were coded and the themes were extracted [36]. The process of coding and decoding was then based on the research questions and the aspects of the SPLISS model. Next, the findings of this study are presented in two parts: the main reasons for introducing sepaktakraw in Taiwan and the strategies Taiwan adopted to rapidly achieve international sepaktakraw success.

## The primary reasons for introducing sepaktakraw in Taiwan

The introduction of the sport of sepaktakraw to Taiwan can be attributed to three main reasons: the low cost of introduction, the breakthrough point of winning medals at the Asian Games, and favourable access to the leadership of international sports organizations.

## Low cost of introduction

There are three reasons for the low cost of developing sepaktakraw in Taiwan: venues, referees, and athlete sources. First, badminton courts are widely available in schools, parks, and community centres and can be used as alternative venues for sepaktakraw; thus, venue infrastructure is a particular advantage. Consequently, the venue rental fee for sepaktakraw promotions or competitions was low. [Interviewee L; P; S]. In fact, when many sports were first introduced to Taiwan, alternative facilities were used to promote their popularization. For example, the venues for promoting Japanese Judo in Taiwan in the early days included ordinary houses and converted barns and factories. In addition, temporary dojos were constructed by laying collapsed rice on roads or open spaces [37]. Second, sepaktakraw referees were borrowed from sports with rules similar to those of sepaktakraw, such as volleyball. A comparison of the rules of volleyball with those of sepaktakraw revealed that they are the same =. However, players are not allowed to touch the ball with their hands, and individuals can touch the ball consecutively [Interviewee R; P; S]. Attracting talent from other sports with similar competition rules is one of the ways other countries have filled the professional refereeing gap quickly and cost effectively. In preparation for the 2022 Beijing Winter Olympics, China had a large shortage of professional judges for ice and snow sports and borrowed and trained people from other sports with similar rules to compensate. For example, the judgement of some turns, jumps and flips in martial arts can be applied to freestyle skiing [38]. The third factor is cross-sport disciplinary selection. In addition to the sport having fewer body size restrictions, talent was recruited directly from sports with similar skills, such as football and badminton [Interviewee L; P; Q; R; S]. Cross-sport disciplinary selection from sports with similar or common skills has succeeded in other countries. For example, many of China's outstanding freestyle skiers and divers, such as Chen Yuxi and Quan Hongchan, were selected from gymnastics, whose skills, such as flipping and turning, are similar to diving [39, 40]. Moreover, China also engaged in cross-sport disciplinary selection from wushu for breakdancing and skateboarding, which had just entered the Olympics, as these events had similar skills of fast footwork, handstands, and flips. [41]. Furthermore, Japan launched the Japan Rising Star Project to carry out large-scale cross-sports discipline selection. Some of the more successful examples include Maho Kakita in competitive cycling and Yoshino Sanju in handball. The former is an experienced football player with good

physical fitness, and the latter is a former player of softball whose throwing skills are similar to those of handball [42].

## The breakthrough point: Medal acquisition in Asian games

Taiwan has always wanted to add new medal-winning events to the Asian Games. After sepaktakraw was listed as an official sport in the 1990 Asian Games, Northeast Asia—China, Japan, and Korea—started to introduce the sport and, within a short period, won many medals in the Asian Games [34]. Because there are fewer competitors and it is easier to achieve results, Mr. Huang Chung-Jen, a Taiwanese entrepreneur, became interested in introducing competitive sepaktakraw when he attended the 2006 Doha Asian Games. After establishing the CTSTF in 2008 and serving as its first president, Mr. Huang Chung-Jen immediately started recruiting individuals for a team to participate in the international sepaktakraw event at the 2014 Asian Games in Incheon, South Korea. This was supported by SAME, with the desire that sepaktakraw would be a breakthrough for Taiwan to win medals at the Asian Games [Interviewee S; [43]]. The use of sepaktakraw as a breakthrough for winning medals at the Asian Games is somewhat consistent with the discussion in Houlihan and Zheng's study of the primary motivations for small countries to invest in elite sports. They point to the need for small countries to utilize and allocate their limited resources more efficiently. Thus, these countries look not only to develop Olympic medal contenders but also to gain access to important global arenas, such as sport national governing bodies (NGOs) and major sporting events [44].

## Favourable access to the leadership of international sports organizations

Compared with other Olympic international sports organizations, the Asian and International Sepaktakraw Federations remain in the early stages of development; therefore, accessing their leadership is easier. Consequently, Taiwan has been actively pursuing leadership positions in international sepaktakraw organizations. Mr. Huang Chung-Jen, the founding CTSTF chairperson, stated that "Taiwan should actively participate in international sports organizations to avoid being marginalized in the international sports arena" in his article "Our National Experience of Active Participation in International Sports Organizations" [45]. Therefore, in addition to establishing the CTSTF in 2008, Mr. Huang actively sought to join the Asian and International Sepaktakraw Federations (ASTAF & ISTAF) as an executive committee member in the same year [Interviewee S; [43]]. Mr. Huang subsequently recommended the second CTSTF president, Mr. Yeh Jeong-Yen, to assume the leadership of international sepaktakraw organizations after his terms as president of the ASTAF and ISTAF ended [Interviewee S; [46]]. In addition to Taiwan, China, Korea and Japan, East Asia emphasizes the need to increase the number of leadership positions in international sports organisations in its national sports policy document. This is closely related to the need to be a part of the development of international sports, to host international competitions, to change the rules of competition, and to decide on the dispatch of referees [47–49].

## Taiwan's strategies to rapidly achieve international sepaktakraw success

Using the nine pillars of the SPLISS model proposed by De Bosscher et al. [12] as an analytical framework, this study summarized the strategies used by Taiwan to rapidly achieve international sepaktakraw competition results. Two aspects, namely, athletic and postcareer support and scientific research, remain unclear. Instead, seven primary areas have been prioritized: financial support, an integrated approach to policy development, foundation and participation, talent identification and development systems, training facilities, coaching provision and

**Table 3. Summary of the case results.**

| Order | Pillars | Case results |
|---|---|---|
| 1 | Financial support | • Funding support from the government |
| 2 | Integrated approach to policy development | • Institutionalization of organisational decision-making |
| 3 | Foundation and participation | • Taking universities as a breakthrough point and gradual shifting towards popularization at the grassroots level |
| 4 | Identification and development talents system | • Selection system based on tournament results |
| 5 | Athletic and postcareer support | • N/A |
| 6 | Training facilities | • Utilizing school badminton venues |
| 7 | Coaching provision and coach development | • Enhanced training for sepaktakraw coaches |
| 8 | (Inter)national competition | • Actively participating in and striving to host international competitions |
| 9 | Scientific research | • N/A |

coach development, and (inter)national competition. Table 3 summarizes the case results of the SPLISS model.

## Funding support from the government

The funding sources for introducing sepaktakraw in Taiwan include government grants and self-financing by the CTSTF. In the early stages, most of the self-funding came from the personal donations of the CTSTF's president [Interviewee C; R; S]. Fig 2 (below) shows the annual budget from 2009 to 2021.

Fig 2 shows that in the early days of sepaktakraw in Taiwan, overall funding strongly correlated with team performance in international competitions. After sepaktakraw was introduced in Taiwan in 2008, the CTSTF began to actively train both men's and women's teams to

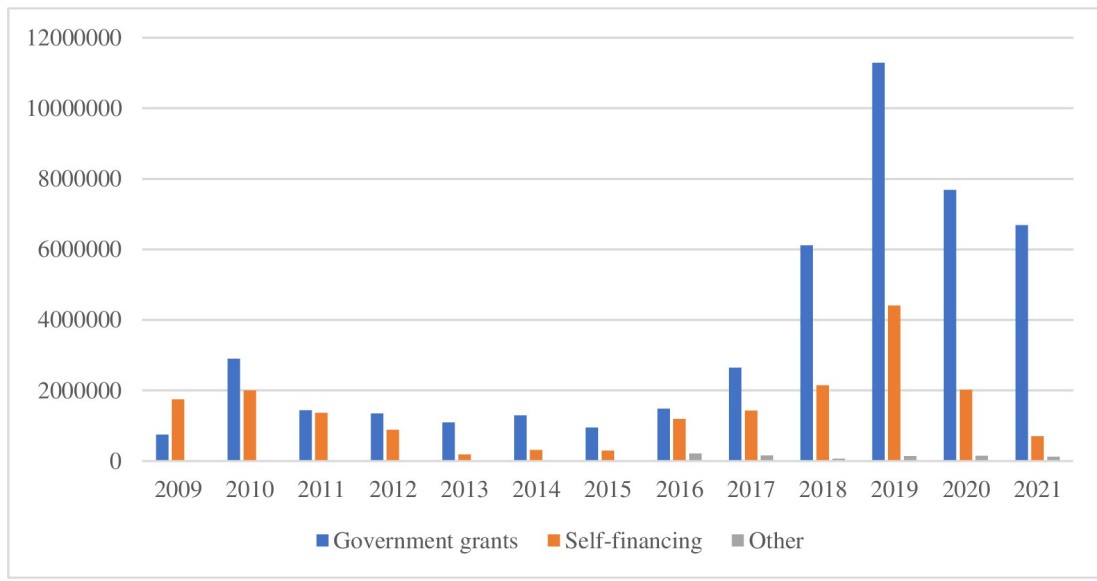

**Fig 2. Annual budget of the CTSTF (2009–2021).** Sources: The annual budgets for the 2009–2015 period were obtained from the CTSTF's internal financial reports, with the permission of the data holder (CTSTF) to use and modify them. The budgets for the 2016–2021 period were obtained from the financial section of the official CTSTF website.

participate in international competitions, such as the World Championships of Sepaktakraw, and proposed a training program in anticipation of participating in the 2014 Incheon Asian Games in Korea. In 2009, the World Games were held in Taiwan, and the ISTAF was funded, including sepaktakraw as a performance event. Therefore, the CTSTF's self-funding in 2009 was greater than the government's subsidy. The former president of the CTSTF and its administrator claimed that "since the 2009 World Games, the sport of sepaktakraw has received considerable attention in Taiwan, which has led to an increase in government subsidies compared with the CTSTF's self-financing" [Interviewee B; S]. In 2015, the men's team achieved the best results in the history of Taiwan's sepaktakraw team at the World Sepaktakraw Championships (the gold medal in the doubles regu match and the silver medal in the regu match of division 2 in the World Championships) [34]. As a result, government grants and federation self-funding have increased since 2016. As of 2018, a new president reformed the development of sepaktakraw in Taiwan in many aspects, including organizing more domestic tournaments and coaching and refereeing seminars, bidding for international tournaments, and submitting plans to the government for funding to increase the number of young sepaktakraw players and reorganize the women's team to increase participation in international tournaments. These measures allowed the team to exceed past achievements, such as receiving the gold medal in the men's 4-person match and the silver medal in the regu match at the 2018 World Sepaktakraw Championships of division 1 and tying for fifth in the top eight of the women's doubles regu match at the premier level in the same event. Therefore, since 2018, CTSTF self-funding has increased from 2 to 4 million New Taiwan dollars, and government funding has increased dramatically from 6 million to 12 million New Taiwan dollars, increasing overall funding for sepaktakraw [[50], Interviewee L; R]. After 2019, due to the COVID-19 epidemic, domestic and many foreign sepaktakraw tournaments were not held, resulting in a decline in funding by both the government and the CTSTF. The overall source of sepaktakraw funding in Taiwan is highly dependent on government subsidies and is strongly related to performance in international competitions. However, it is necessary to overcome the overreliance on government financial support for long-term development by enlisting the support of other community forces (such as sponsorships from corporations or donations from social organizations) to sustainably develop sepaktakraw in Taiwan.

## Institutionalization of organisational decision-making

In the early stages, the CTSTF president was in charge of making important decisions. Initially, the CTSTF had few internal members and limited operations, including organizing domestic events and coaching and refereeing seminars, assisting in the training of team players, and participating in international competitions [Interviewee C]. In addition, aspects of the internal organization and functioning of the CTSTF in the early days had to be better developed. Although the CTSTF developed its official website in 2015, the website lacked management and consisted only of competition announcements and a brief introduction to the origins and development of the sepaktakraw movement [Interviewee B; [51]]. Critical decisions, such as the selection of representative teams and participation in international competitions, were almost always made by the CTSTF president at the time [Interviewees B, F, H].

In 2018, the new CTSTF president began to carry out a series of reforms at the organisational level, and the CTSTF's organisational decision-making gradually became institutionalized. In terms of organisational structure, additional ad hoc committees were established, and scholars and experts were invited to establish the Operations Evaluation Committee, the Election Committee, and the Anti-Doping Committee, as well as the Organizing Committee for International Competitions [[52–54]; Interviewee O; R]. In terms of institutional aspects, the

CTSTF has revised its organisational regulations, clarified the qualifications of its members, and confirmed the annual financial accounts and budgets and the annual work plan. At the operational level, Athletics Selection and Training Committee meetings are held regularly to discuss the criteria for selecting national team members, confirming the list of coaches and athletes who will participate in international competitions. In 2020, the CTSTF proposed a medium- and long-term development plan according to the National Sports Law and strengthened its tasks as stipulated in the newly revised regulations [[50]; Interviewee O; R]. Overall, after 2018, the CTSTF began to institutionalize its organisational reforms and decision-making processes. A very important and crucial element of good governance is ensuring that the organization's "decision-making process" and "decision-implementation process" are fair, open, free from favouritism, and comply with laws and regulations, ensuring that the CTSTF will continue to sustainably promote sepaktakraw in Taiwan.

## The use of universities as a catalyst to gradually shift towards popularization at the grassroots level

During the early stages of the CTSTF's efforts to increase participation, the focus was on the departments of physical education in teacher training colleges. National sepaktakraw team athletes were supported by the CTSTF in 2010 to promote sports in teacher colleges for three reasons. First, the intention was to rapidly promote sepaktakraw and provide enough athletes to participate in international competitions. Second, these sponsored athletes could coach sepaktakraw clubs and school teams in nearby primary and secondary schools during their free time, in addition to organizing and competing in high-level competitions. Third, these sponsored athletes were expected to have careers as teachers or coaches at elementary, junior, and senior high schools after completing their studies. In addition to teaching sepaktakraw in schools, they also organized teams to participate in CTSTF tournaments. Notably, a significant number of participants were drawn from secondary school sports teams. Many junior high and elementary schools invited badminton and football teams to participate in sepaktakraw tournaments, and the CTSTF arranged for national players to serve as coaches in those schools' clubs or teams [Interviewee B; F; H]. At this stage, the CTSTF focused its development on the elementary, junior high, and senior high school levels to popularize sepaktakraw in Taiwan.

In recent years, the federation has begun investing in organizing more youth-level tournaments, the performance results of which can be used for high school or university entrance examinations. The federation has also begun broadcasting tournaments and the prize fund mechanism, attracting many schools below the tertiary level to participate in tournaments [Interviewee L; O; R]. For this purpose, in addition to continuing to increase the number of ready-to-play athletes from the tertiary level who participate in international competitions, there has been a return to the grassroots level of development to achieve more sustainable breakthroughs and develop future athletic talent. Numerous interviewees reported that the participation rate for high schools in the national sepaktakraw tournament in 2023 was six times higher than that in 2016 and 12 times higher than that for elementary schools [Interviewee H; Q]. Sepaktakraw in Taiwan has achieved significant results in terms of participants and has been gradually rooted downwards. The low birth rate, however, may make it difficult to recruit and compete with other sports associations for the scarce pool of young athletes in Taiwan.

## Selection system based on tournament results

When sepaktakraw was first introduced in Taiwan, there were no sepaktakraw players; thus, players were recruited from school badminton, football, and taekwondo teams. After

participating in domestic competitions and short training programs, the players were soon assigned to participate in international tournaments [Interviewee E; B]. From 20082016, 12 team players have attended the World Sepaktakraw Championships each year. Nine places were released to players who had performed well in the domestic championships with the endorsement of the CTSTF president and the team coaches. The CTSTF president used the remaining places to allow school players who wished to develop their sepaktakraw skills to attend international competitions [Interviewee F; J]. The new president came into office in 2018, and he applied the talent selection and training model previously used to develop badminton in Taiwan to develop sepaktakraw. For the first time, in 2019, some domestic tournaments were converted into domestic sepaktakraw ranking tournaments, and existing players were divided into Groups A and B. The threshold for Group A players was participating in the World Sepaktakraw Championships more than twice, with subsequent players advancing by obtaining excellent performance in the ranking tournaments. After being promoted to Group A, if a player had obtained a high number of points (3 points for the champion, 2 points for the first runner-up, and 1 point for the second runner-up) in the preinternational cup matches compared with other players, and according to the comprehensive evaluation by the selection committee, an athlete was selected as a national team member [[54], Interviewee L; O; Q; R]. The badminton talent selection and training model adopted by Taiwanese sepaktakraw has helped Taiwan achieve favourable results in international tournaments. This is due to its emphasis on competition and fairness. However, relying solely on the annual ranking tournament may incentivize elite players to miss team training because of poor match conditions on the day of the tournament. Therefore, selecting and training outstanding players in the sport is a complex but critical task for Taiwanese sepaktakraw. This selection and training must be done while keeping fair competition in mind and not missing out on potential players.

## Utilizing school badminton venues

In the early stages, when the Federation did not have sufficient funds, the teams relied more on school stadiums or county and city stadiums for training. To save money on the long-term rental of venues, the Federation used the president's connections to coordinate with colleges and universities or with county and city governments to provide accessible venues for team training before international tournaments. However, most venues and equipment utilized badminton courts [Interviewee B; F]. With the new president in office since 2018, the AP centre, a specialized venue used by professional badminton players, has been designated a training venue for the team. In preparation for the sepaktakraw event at the 2023 Asian Games in Hangzhou, the team used the badminton court at the East Training Base National Sports Training Center at the National Taitung University High School of Physical Education and Sports [Interviewee L; R]. When not training as a representative team, players relied extensively on badminton courts in their school gymnasiums, such as those at Chia Nan University of Pharmacy and Science [Interviewee O; N]. Taiwan's facilities have seen the sport of sepaktakraw gradually develop from the early days of using borrowed venues to dedicated badminton venues. However, Taiwan still must address the complex issue of the lack of dedicated sepaktakraw venues.

## Enhanced training for sepaktakraw coaches

In the early days, most team members were from the same school, and if the school team had a coach, the coach was invited by the president of the federation to coach the national team and lead it in international competitions [Interviewee C]. When the Federation initially organized coaching workshops, there was a severe shortage of sepaktakraw-specialized lecturers,

resulting in poor sepaktakraw coach training [Interviewees B; E]. Since the new president took office in 2018, the training of coaches has strengthened. As of 2022, 314 Grade C coaches, 49 Grade B coaches, and 6 Grade A coaches have been trained under the Taiwan government's single sport coaching licence grading system [55]. Among the current six Grade A coaches, three take turns providing training for the national men's and women's teams and leading the teams in international competitions. As a result, the national team's training conditions have improved greatly since it was first established [Interviewee L]. The lecturers of coaching seminars have begun to be taught by senior sepaktakraw sports team players or coaches, and the quality of the training has improved. However, there remains an urgent need for a breakthrough in the quality of coaching training, such as hiring international coaches to conduct coaching lectures or sending coaches abroad for training [Interviewee O; R].

### Actively participating in and striving to host international competitions

In the early days, when the federation was short of funds and not competitive, bidding for or competing in international competitions was not prioritized [Interviewees H; B; A]. In contrast, the Federation was more interested in developing domestic competitions at that time. To increase the number of participants and attract resources from counties and municipalities, the Federation sought to include sepaktakraw as an official sport in future National Games or National Secondary School Games [Interviewee C; F]. Following the new president's inauguration in 2018, he began to actively bid for or organize international sepaktakraw tournaments, such as the 2019 YONEX Taiwan International Sepaktakraw Invitational Championships and the 2019 Asian Junior (U19) Sepaktakraw Championships, where national youth teams from Asian countries, such as Malaysia, Thailand, India, Vietnam, Japan, South Korea, and the Philippines, were invited to compete in Taiwan [56–58]. These accomplishments led to his election as Vice President and Executive Committee member of the ASTAF at the 2021 General Assembly [59] and as Vice President of the ISTAF at the 2022 General Assembly [60]. In addition, to qualify for the Asian Games, the CTSTF has been actively sending teams to participate in various international sepaktakraw tournaments in recent years, such as the Asia Sepaktakraw Championship [Interviewee O; R]. Currently, Taiwan has bid for, hosted, and participated in international sepaktakraw tournaments. Taiwan has benefited from this by having a voice in international sepaktakraw organizations and achieving positive results in international tournaments. For the sport to be sustainable in Taiwan, the SAME should include it in the National Games and Secondary School Games. By offering players scholarships and incentivizing involvement in the sport in higher education, better young players can be encouraged to participate in national team selection and training.

### Discussion

In our investigation of why Taiwan introduced the emerging competitive sport of sepaktakraw and the strategies used to quickly achieve international sporting success, we identified a few relevant issues for analysis:

1. Whether the introduction of the emerging competitive sport by the state was accidental or intentional

2. The trajectory of the state's development after the introduction of the emerging sport

3. How the lack of athletes participating in international competitions during the introduction of the emerging sport was addressed

4. The sustainable development of newly introduced sports

## Contingency and inevitability of the state introducing a new competitive sport

Sepaktakraw was introduced in Taiwan in the 1980s by Southeast Asian labourers as a leisure and entertainment activity, but it did not gain much traction. Competitive sepaktakraw did not begin to develop in Taiwan until the Taiwanese entrepreneur Mr. Huang Chung-Jen visited the 2006 Doha Asian Games and subsequently actively set up the CTSTF and served as its first president, donating money and making efforts to form a team to participate in the Asian Games. From this perspective, the introduction of sepaktakraw to Taiwan seems to be deliberate.

However, since 1990, when sepaktakraw was recognized as an official sport of the Asian Games, the Taiwanese government has been trying increase medal wins in the Asian and Olympic games; sepaktakraw, kabaddi and other sports included in the Asian Games have naturally attracted the attention of entrepreneurs and the government. In particular, sepaktakraw not only serves as a means for Taiwan to win medals in the Asian Games but also has the advantages of being low cost and low barrier for entry into international sports organizations. Therefore, when Mr. Huang Chung-Jen expressed his intention of introducing and forming a team to participate in the Asian Games, the government immediately responded and gradually increased its support for developing competitive sepaktakraw. Using sepaktakraw as a breakthrough point for the Asian Games is, to some extent, consistent with the findings of Houlihan and Zheng's study on the primary motivations behind small countries' investments in competitive sports. The authors noted that small countries with limited resources must use and deploy their resources more efficiently in the hope of increasing the number of Olympic medal contenders. More pragmatically, this approach also allows them to enter important global fields, such as sports NGOs and megasports events [44]. In this context, especially for small countries, the introduction of emerging competitive sports such as sepaktakraw to prominence in the international competitive sports arena is intentional. The development of niche sports in some small countries or regions has also achieved good results worldwide, such as Samoa, Tonga and New Zealand in men's rugby unions and Slovenia and Croatia in men's handball [44]. In fact, some influential competitive sports countries have similar practices. Zeng and Chen noted that China's continued success in the Olympic Games has been due to prioritizing resources in specific sports, such as diving and table tennis, which are less popular globally and less competitive internationally [61]. Like China, at the recently concluded Paris 2024 Olympic Games, Japan won multiple gold medals in specific sports that fewer countries have invested in developing and participating in, such as men's and women's skateboarding [62].

## Developmental trajectory of a new competitive sport by the state

Since the disintegration of the Soviet Union, countries worldwide have gradually extended their military and political competition to competitive sports competition. The competition has become increasingly fierce, and "the global sporting arms race" has taken shape [13]. Under these circumstances, introducing emerging Asian and Olympic sports and competing for medals in Asian and Olympic sporting events have become critical issues. From the developmental trajectory of the introduction of competitive sepaktakraw to Taiwan, we understand the importance of resource supply and system establishment. In terms of resources, this covers basic elements such as funds, venues, and talent (players, coaches, referees). When Taiwan introduced competitive sepaktakraw, it relied heavily on the government to provide these infrastructure resources, especially funding subsidies from the SAME; the provision of training and competition venues in various counties, municipalities, and schools at all levels; and sport-related university talent resources (e.g., players, coaches, and referees). The institutional

establishment covers organizations, competitions, talent selection, training, and other elements. In Taiwan's case, since the new CTSTF president took office in 2018, many changes have occurred, including reorganizing and refinement of the organisational structure, hosting and competing in domestic and foreign events, selecting and training talented individuals, and providing specialized training for coaches and referees. Establishing an institutionalized system will lay a foundation for the sustainable development of competitive sepaktakraw in Taiwan. Perhaps when a country introduces new competitive sports, it will inevitably go through a "first have, then refine" development trajectory. We must first identify and develop certain infrastructures (funding, venues, talent) and then pursue establishing an enhanced system (organization, competitions, talent selection, training).

## Using a cross-sports discipline talent selection strategy

To solve the problem of selecting and participating in international competitions in the early stages of competitive sepaktakraw, Taiwan directly selected or recruited players from sports requiring similar skills (such as football and badminton) to participate in the events. This approach was not unusual because past research has also revealed that when a newly developed competitive sport is introduced, players with similar sports skills are usually selected to train and participate in international competitions to solve the problem of player shortages. For example, when Japanese judo first spread to other countries, local combat sports players, such as wrestlers, were the first to participate in judo training and events [63]. Similarly, after Norway and Germany, China ranked third in the 2022 Winter Olympics. In addition to the host country's advantages, selecting talent across disciplines is one key to success. For example, Xu Meng-Tao, a Chinese freestyle skiing aerial gold medallist in the Winter Olympic Games, noted that her early gymnastics training had a transfer effect on her freestyle skiing aerial training [64].

## Sustainable development issue concerning the introduction of a new competitive sport

As mentioned, Taiwan's sepaktakraw development strategy prioritizes the seven key factors in the SPLISS model for the success of competitive sports. For example, universities are used as a means to extend sports to the grassroots level and rely heavily on school badminton venues. However, development strategies for athletic and postcareer support and scientific research are still lacking. It was hoped that Taiwan would make rapid progress in the early phase of spaktakraw introduction. However, Taiwan also the problem of limited resources. Therefore, its strategies focused on the necessary conditions for developing competitive sports (i.e., financial support) and the initial elements of the development of competitive athletes (i.e., talent identification and development systems and training facilities) rather than prioritizing the subsequent aspects of competitive athletic performance (i.e., athletic and postcareer support and sports science) [12]. From this perspective, we can see that, except for athletes' careers, postcareer support, and scientific research, almost all aspects have entered a relatively prosperous developmental stage, but there remains a long way to go before success and sustainable development are achieved. In other words, Taiwan should have other sources of funding, full institutionalization and implementation of organisational decisions, greater numbers of people participating in sepaktakraw, comprehensive selection mechanisms, a wide range of dedicated venues, higher-quality training systems for coaches, and official competition in the National Games at different levels. Continuing to strengthen the use of the SPLISS model to review and revise the sepaktakraw development strategy may be necessary for Taiwan's future successful and sustainable development of competitive sepaktakraw. For other countries that want to

introduce or have recently introduced emerging competitive sports, Taiwan's experience in developing competitive sepaktakraw may serve as a reference.

## Conclusions

This study explored the reasons and strategies for introducing and developing sepaktakraw in Taiwan and essential related issues. This study revealed the primary reasons for the introduction of competitive sepaktakraw to Taiwan: the lower cost and greater feasibility of introducing competitive sepaktakraw, making it easier to win medals in the Asian Games and facilitating Taiwan's entry into the sports leadership of international organizations. Taiwan's sepaktakraw development strategy prioritizes seven aspects: (1) obtaining government financial support; (2) institutionalizing organisational decision-making; (3) using universities as a catalyst to and gradually shifting towards popularization at the grassroots level; (4) selecting players on the basis of competition results; (5) using school badminton courts; (6) strengthening coach training; and (7) actively participating in and striving to host international tournaments. Four main issues merit attention: (1) the contingency and intentionality of introducing a new competitive sport by the state; (2) the developmental trajectory of a newly introduced competitive sport by the state; (3) the use of a cross-sports talent selection strategy; and (4) the sustainable development of a new competitive sport.

In the increasingly fierce "global sporting arms race," how to create new shortcuts for developing competitive sports has become increasingly important. The reasons and strategies for introducing and developing sepaktakraw in Taiwan may provide a reference for existing competitive sports research and related policy formulation and implementation groups in other countries. When introducing emerging competitive sports at the national level, more references to policy experiences, innovative policy ideas and new shortcuts for developing competitive sports are feasible.

## Author Contributions

**Conceptualization:** Ding-Yi Wu, Tien-Chin Tan, Yan-Ting Wang.

**Data curation:** Ding-Yi Wu.

**Formal analysis:** Ding-Yi Wu.

**Investigation:** Ding-Yi Wu.

**Methodology:** Ding-Yi Wu, Tien-Chin Tan.

**Visualization:** Ding-Yi Wu.

**Writing – original draft:** Ding-Yi Wu.

**Writing – review & editing:** Tien-Chin Tan, Yan-Ting Wang.

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
