## [Decision Letter · Decision Letter 0]

19 Jul 2024

PONE-D-24-14953A new shortcut for competitive sports development? The purpose and strategy of developing and introducing sepaktakraw in TaiwanPLOS ONE

Dear Dr. Tan,

Thank you for submitting your manuscript to PLOS ONE. After careful consideration, we feel that it has merit but does not fully meet PLOS ONE’s publication criteria as it currently stands. Therefore, we invite you to submit a revised version of the manuscript that addresses the points raised during the review process.

Please carefully read the review comments and respond accordingly. 

We look forward to receiving your revised manuscript.

Kind regards,

Yue Gong

Academic Editor

PLOS ONE

Journal Requirements:

2. In the online submission form, you indicated that "The datasets generated and analyzed during the current study are available from the corresponding authors upon reasonable request."

Reviewers' comments:

Reviewer's Responses to Questions

**Comments to the Author**

1. Is the manuscript technically sound, and do the data support the conclusions?

Reviewer #1: Yes

Reviewer #2: Yes

2. Has the statistical analysis been performed appropriately and rigorously? 

Reviewer #1: N/A

Reviewer #2: Yes

3. Have the authors made all data underlying the findings in their manuscript fully available?

Reviewer #1: No

Reviewer #2: Yes

4. Is the manuscript presented in an intelligible fashion and written in standard English?

Reviewer #1: Yes

Reviewer #2: Yes

5. Review Comments to the Author

Reviewer #1: This manuscript investigated the introduction and development of a new sport, sepaktakraw, in Taiwan. The SPLISS model was appropriately used. This study provided detailed insights as several key members of the sport were interviewed. This study also identified a major reason for the local government to support this sport, ie, winning medals in international competitions. This aspect is different from many western countries.

Several minor points need to be addressed.

1. The sentence ‘after the new president took office in 2018’ has been used multiple times throughout the manuscript. Please modify or simply.

2. Line 124: Chinese Taipei ‘Sepak Takraw’ Federation or Chinese Taipei ‘SepakTakraw’ Federation? Also in line 183-184.

3. Line 133: Sports Administration, Ministry of Education.

4. Line 218: In 2010, sepaktakraw received widespread attention. Please be more specific on ‘widespread attention’, such as the number of participating countries or athletes.

5. Line 230-231: 2018 World Sepaktakraw Championships of ‘division’ 1…?

6. Line 233: …government funding has increased dramatically. Please be more specific on the number of funding.

7. Line 234-236: The best international competition result was achieved in 2015, but the government budget peaked in 2019 and was decreasing since. Please clarify.

8. Line 268: sending athletes to PE departments and serving as club coaches seem to be unrelated. Please clarify.

9. Line 270: …ensure these athletes will become PE teachers…Attending normal universities does not guarantee these athletes will become PE teachers. Please revise.

10. Line 283-285: please be more specific on which year(s) was ‘now’ and ‘early years’.

11. Line 351: …countries were widely invited to compete in Taiwan. Please be more specific on the number of countries invited.

12. Line 384: …Mr. ‘Huang’ Chung-Jen…

13. Figure 1: the background and numbers in Y-axis can be modified according to the Journal’s style.

Reviewer #2: Dear Author！

After careful review of this article. Read through, this article explores why Taiwan introduced an emerging sport, rattan ball (sepaktakraw), and how it achieved international sporting success in the short term. The study uses the sport policy model proposed by De Bosscher et al. as a framework for analysis. Using official government and sports federation documents, media reports, and semi-structured interviews with 18 key stakeholders, the study identified three main reasons for the introduction of rattanball: (1) low cost of introduction; (2) medal breakthrough at the Asian Games; and (3) facilitating access to the leadership of international sports organizations. The development strategy of rattan in Taiwan includes seven aspects: obtaining government financial support, institutionalizing organizational decision-making, using universities as a breakthrough point and gradually expanding to the grassroots level, selecting players based on tournament results, utilizing school badminton courts, enhancing coach training, and actively participating in and striving for the hosting of international tournaments. In addition, the study raises four concerns: the contingency and inevitability of the introduction of new competitive sports, the trajectory of development, the strategy of cross-sport talent selection, and the issue of sustainability. However, the article's argumentation process is too uncritical and there are areas that need to be revised.

1.Clarity and conciseness: in the abstract, improve scholarship.

Abstract required before introduction section also required to write properly. It should be written in the following steps;

a) Write the concept as per your paper title (2 lines minimum).

b) Objective (2 lines)

c) Tools (2 lines)

d) Output (3 lines)

e) Output with application in real life or in the industry (mandatory)

2.Deepening of the theoretical framework: to consider further generalising and expanding the existing theoretical framework, it is recommended that a more in-depth literature review of existing research be conducted to highlight the importance and innovation of this study in the current field of academic research.

In this section,

a) the background of this research domain is way sufficing and the justification for this research (add some proper implication fig./picture in the background)

b) research gap, weakly presented.

c) the proposal of this research does not seem to be clearly presented to address the research gap. Please revise the Introduction section accordingly.

d) Literature review shall be comprehensive (rather than brief) to discuss the right breadth of knowledge and recent works in the area.

e) Better to add more recent work in the contribution table.

f) Author should provide the real case study with an appropriate diagram/picture to explain with details.

g) Managerial implications would be useful.

In conjunction with analysing articles in related fields, you can refer to relevant literature on public participation and public complaints to enhance the theoretical support of the introductory section, highlight the theoretical orientation, focus on the argument from a theoretical perspective, and put forward relevant theoretical support. The following articles can be highlighted for reference:

①Di, K., Chen, W., Shi, Q., Cai, Q., & Liu, S. (2024). Analysing the impact of coupled domestic demand dynamics of green and low-carbon consumption in the market based on SEM-ANN. Journal of Retailing and Consumer Services, 79, 103856.

②Di, K., Chen, W., Zhang, X., Shi, Q., Cai, Q., Li, D., ... & Di, Z. (2023). Regional unevenness and synergy of carbon emission reduction in China's green low-carbon circular economy. Journal of Cleaner Production, 420, 138436.

③Di, K., Chen, W., Shi, Q., Cai, Q., & Zhang, B. (2024). Digital empowerment and win-win co-operation for green and low-carbon industrial development: Analysis of regional differences based on GMM-ANN intelligence models. Journal of Cleaner Production, 141332.

3.Clarity and Structure:The article is well-structured, with a clear introduction, methodology, results, and discussion sections. However, some parts could benefit from more explicit transitions between sections to improve the overall flow.

4.Literature Review:While the study adopts De Bosscher et al.'s model as the analytical framework, a more detailed explanation of this model and its relevance to sepaktakraw's introduction in Taiwan would strengthen the theoretical foundation.

5.Data Collection and Analysis:The use of semistructured interviews with 18 key stakeholders is appropriate, but more information on the selection criteria for these stakeholders and the interview process would enhance the credibility of the findings. Additionally, detailing how media reports and official documents were analyzed could provide more transparency.

6.Findings and Discussion:The three main reasons for introducing sepaktakraw in Taiwan are clearly stated, but providing more context or examples for each reason would make the findings more compelling. For instance, elaborating on the specific costs saved, or success stories of other countries in similar positions, would add depth.

7.Development Strategy:The seven prioritized aspects of the development strategy are comprehensive. However, the article could benefit from discussing potential challenges or limitations associated with each aspect. For instance, the practical difficulties in using school badminton courts or obtaining sustained government financial support could be addressed.

8.Issues of Concern:The four issues of concern raised are critical and thought-provoking. To enhance the discussion, incorporating comparative examples from other countries that have introduced new sports could provide a broader perspective on the contingency and inevitability, developmental trajectory, cross-sports talent selection strategy, and sustainable development issues.

6. PLOS authors have the option to publish the peer review history of their article (what does this mean?). If published, this will include your full peer review and any attached files.

Reviewer #1: No

Reviewer #2: No

---

## [Author Response · Author response to Decision Letter 0]

17 Sep 2024

Dear Reviewers,

Thank you so much for your letter.

A statement about the openness and accessibility of our research data has been reintroduced. Our apologies for any misunderstandings about this statement in the past. We thought we should copy the example sentence in this section regarding our case. Our intention is to include all relevant data within our manuscript.

Sincerely,

Tien-Chin Tan

---

## [Decision Letter · Decision Letter 1]

6 Nov 2024

PONE-D-24-14953R1A new shortcut for competitive sports development? The purpose and strategy of developing and introducing sepaktakraw in TaiwanPLOS ONE

Dear Dr. Tan,

Thank you for submitting your manuscript to PLOS ONE. After careful consideration, we feel that it has merit but does not fully meet PLOS ONE’s publication criteria as it currently stands. Therefore, we invite you to submit a revised version of the manuscript that addresses the points raised during the review process.

Please respond to Reviewer 2 comments and return your revision on time.

We look forward to receiving your revised manuscript.

Kind regards,

Yue Gong

Academic Editor

PLOS ONE

Journal Requirements:

Reviewers' comments:

Reviewer's Responses to Questions

**Comments to the Author**

1. If the authors have adequately addressed your comments raised in a previous round of review and you feel that this manuscript is now acceptable for publication, you may indicate that here to bypass the “Comments to the Author” section, enter your conflict of interest statement in the “Confidential to Editor” section, and submit your "Accept" recommendation.

Reviewer #1: All comments have been addressed

Reviewer #2: All comments have been addressed

2. Is the manuscript technically sound, and do the data support the conclusions?

Reviewer #1: Yes

Reviewer #2: Yes

3. Has the statistical analysis been performed appropriately and rigorously? 

Reviewer #1: N/A

Reviewer #2: Yes

4. Have the authors made all data underlying the findings in their manuscript fully available?

Reviewer #1: Yes

Reviewer #2: Yes

5. Is the manuscript presented in an intelligible fashion and written in standard English?

Reviewer #1: Yes

Reviewer #2: Yes

6. Review Comments to the Author

Reviewer #1: The authors have answered all of my comments. There is no further comment. The authors have answered all of my comments. There is no further comment.

Reviewer #2: Dear Author！

After a careful review of this article. After reading through the article, the argumentative process is too uncritical and there are areas that need to be revised.

1... The deepening of the theoretical framework is an important part of ensuring that the research is scholarly and theoretical. In further generalizing and expanding the existing theoretical framework, a more in-depth literature review of the existing research is needed to highlight the importance and innovation of this study in the current field of academic research. To further provide the theoretical and scholarly nature, a more in-depth literature review of existing studies is recommended to highlight the importance and innovations of this study in the current field of scholarly research.

. Combined with the analysis of articles in related fields, it can refer to the relevant literature on public participation and public complaints to enhance the theoretical support of the introductory section, highlight the theoretical orientation, focus on the argument from the theoretical perspective, and present the relevant theoretical support. The following articles can be highlighted for reference:

①Sun, T., Di, K., & Shi, Q. (2024). Digital economy and carbon emission: The coupling effects of the economy in Qinghai region of China. Heliyon, 10(4).

②Sun, T., Di, K., Shi, Q., Hu, J., & Zhang, X. (2024). Study on the development path of low-carbon retail clusters empowered by digital empowerment. Journal of Retailing and Consumer Services, 81, 104006.

③Wang, J., Qiao, L., Zhu, G., Di, K., & Zhang, X. (2025). Research on the driving factors and impact mechanisms of green new quality productive forces in high-tech retail enterprises under China's Dual Carbon Goals. Journal of Retailing and Consumer Services, 82, 104092.

④Xue, H., Cai, M., Liu, B., Di, K., & Hu, J. Sustainable development through digital innovation: Unveiling the impact of big data comprehensive experimental zones on energy utilization efficiency. Sustainable Development.

⑤Liu, S., Cai, Q., Wang, M., & Di, K. (2024). Urban public services and fertility intentions of internal migrants in China. Plos one, 19(3), e0300345.

⑥Cai, Q., Chen, W., Wang, M., & Di, K. How Does Green Finance Influence Carbon Emission Intensity? A Non-Linear fsQCA-ANN Approach. Polish Journal of Environmental Studies.

2.To increase the framework diagram of this article to enhance the readability and wholeness of the level of the article. In order to enhance the readability and wholeness of the level of the article, the framework diagram of this paper can be added. The framework diagram clearly displays the existing theoretical framework, the extension and innovation points of this study, and the research methodology, which helps readers to understand the study more intuitively. The addition of such diagrams not only improves the visual effect of the article, but also makes the complex theoretical framework easier to understand.

3. Strengthen the revision of academic language to improve the rigor and scientificity of academic papers. In order to improve the rigor and scientificity of academic papers, it is necessary to strengthen the revision of academic language. The academic level of the paper can be improved by refining the expression, ensuring the accuracy of terminology and logical rigor. Avoiding colloquial expressions and using appropriate academic terminology and structured sentences can help highlight the theoretical and academic nature of the research. In addition, maintaining a consistent academic style and standardized citation are key to enhancing the quality of the paper.

7. PLOS authors have the option to publish the peer review history of their article (what does this mean?). If published, this will include your full peer review and any attached files.

Reviewer #1: No

Reviewer #2: No

---

## [Author Response · Author response to Decision Letter 1]

21 Nov 2024

Dear Reviewer 2, 

Thanks for your valuable modifications suggestions. We have already amended the article based on your suggestions. The following are our responses and amendments to reviewer's comments.

Q1: The deepening of the theoretical framework is an important part of ensuring that the research is scholarly and theoretical. In further generalizing and expanding the existing theoretical framework, a more in-depth literature review of the existing research is needed to highlight the importance and innovation of this study in the current field of academic research. To further provide the theoretical and scholarly nature, a more in-depth literature review of existing studies is recommended to highlight the importance and innovations of this study in the current field of scholarly research.

Answer 1: Your suggestions are greatly appreciated. To provide a deeper understanding of the existing theoretical framework, we have re-examined the relevant literature and added relevant content to the literature review section. Corrections can be found on lines 114-122.

Q2: Combined with the analysis of articles in related fields, it can refer to the relevant literature on public participation and public complaints to enhance the theoretical support of the introductory section, highlight the theoretical orientation, focus on the argument from the theoretical perspective, and present the relevant theoretical support. The following articles can be highlighted for reference.

Answer 2:Thanks for your suggestions. Following the reviewer’s recommendation, we have included citations in the introduction to articles in the relevant fields and rediscussed the research gaps. For changes, please refer to lines 50-56.

Q3: To increase the framework diagram of this article to enhance the readability and wholeness of the level of the article. In order to enhance the readability and wholeness of the level of the article, the framework diagram of this paper can be added. The framework diagram clearly displays the existing theoretical framework, the extension and innovation points of this study, and the research methodology, which helps readers to understand the study more intuitively. The addition of such diagrams not only improves the visual effect of the article, but also makes the complex theoretical framework easier to understand.

Answer 3: Thank you very much for your suggestions. To improve the visual appeal of the article, we added a diagram of the framework of this article to the introduction section as suggested by the reviewers (Table 1). The nine pillars of the SPLISS model and a summary of the case results (Table 3) have been added to the literature review and findings sections. Please refer to lines 64-68, 101-102, and 318-322 for corrections.

Q4: Strengthen the revision of academic language to improve the rigor and scientificity of academic papers. In order to improve the rigor and scientificity of academic papers, it is necessary to strengthen the revision of academic language. The academic level of the paper can be improved by refining the expression, ensuring the accuracy of terminology and logical rigor. Avoiding colloquial expressions and using appropriate academic terminology and structured sentences can help highlight the theoretical and academic nature of the research. In addition, maintaining a consistent academic style and standardized citation are key to enhancing the quality of the paper.

Answer 4: Thank you for the suggestion. We have amended the article to make it more academic by re-examining the terminology throughout. The article has also been reviewed by an academic editing company for accuracy, scholarship, and consistency in terminology and citations.

We look forward to hearing from you soon!

Yours Sincerely, 

Tien-Chin Tan

---

## [Decision Letter · Decision Letter 2]

29 Dec 2024

A new shortcut for competitive sports development? The purpose of and strategy for developing and introducing sepaktakraw in Taiwan

PONE-D-24-14953R2

Dear Dr. Tan,

We’re pleased to inform you that your manuscript has been judged scientifically suitable for publication and will be formally accepted for publication once it meets all outstanding technical requirements.

Kind regards,

Yue Gong

Academic Editor

PLOS ONE

Additional Editor Comments (optional):

Reviewers' comments:

Reviewer's Responses to Questions

**Comments to the Author**

1. If the authors have adequately addressed your comments raised in a previous round of review and you feel that this manuscript is now acceptable for publication, you may indicate that here to bypass the “Comments to the Author” section, enter your conflict of interest statement in the “Confidential to Editor” section, and submit your "Accept" recommendation.

Reviewer #2: All comments have been addressed

2. Is the manuscript technically sound, and do the data support the conclusions?

Reviewer #2: Yes

3. Has the statistical analysis been performed appropriately and rigorously? 

Reviewer #2: Yes

4. Have the authors made all data underlying the findings in their manuscript fully available?

Reviewer #2: Yes

5. Is the manuscript presented in an intelligible fashion and written in standard English?

Reviewer #2: Yes

6. Review Comments to the Author

Reviewer #2: It has been modified to be very good and agreed to be published。Overall, the content is detailed and scientifically rigorous

7. PLOS authors have the option to publish the peer review history of their article (what does this mean?). If published, this will include your full peer review and any attached files.

Reviewer #2: No

---

## [Editor Report · Acceptance letter]

2 Jan 2025

PONE-D-24-14953R2 

PLOS ONE

Dear Dr. Tan, 

I'm pleased to inform you that your manuscript has been deemed suitable for publication in PLOS ONE. Congratulations! Your manuscript is now being handed over to our production team.

Kind regards, 

on behalf of

Dr. Yue Gong 

Academic Editor

PLOS ONE